# Improved Streamflow Forecast in a Small-Medium Sized River Basin with Coupled WRF and WRF-Hydro: Effects of Radar Data Assimilation

**Tianwei Gu** , **Yaodeng Chen, Yufang Gao \*, Luyao Qin, Yuqing Wu and Yazhen Wu**

Key Laboratory of Meteorological Disaster of Ministry of Education (KLME), Joint International Research Laboratory of Climate and Environment Change (ILCEC), Collaborative Innovation Center on Forecast and Evaluation of Meteorological Disasters, School of Applied Meteorology, Nanjing University of Information Science and Technology, Nanjing 210044, China; 20181201023@nuist.edu.cn (T.G.); keyu@nuist.edu.cn (Y.C.); qinly@nuist.edu.cn (L.Q.); 20181208038@nuist.edu.cn (Y.W.); 20191208028@nuist.edu.cn (Y.W.)
\* Correspondence: gaoyf@nuist.edu.cn

**Abstract:** Accurate and long leading time flood forecasting is very important for flood disaster mitigation. It is an effective method to couple the Quantitative Precipitation Forecast (QPF) products provided by Numerical Weather Prediction (NWP) models to a distributed hydrological model with the goal of extending the leading time for flood forecasting. However, the QPF products contain a certain degree of uncertainty and would affect the accuracy of flood forecasting, especially in the mountainous regions. Radar data assimilation plays an important role in improving the quality of QPF and further improves flood forecasting. In this paper, radar data assimilation was applied in order to construct a high-resolution atmospheric-hydrological coupling model based on the WRF and WRF-Hydro models. Four experiments with conventional observational and radar data assimilation were conducted to evaluate the flood forecasting capability of this coupled model in a small-medium sized basin based on eight typical flood events. The results show that the flood forecast skills are highly QPF-dependent. The QPF from the WRF model is improved by assimilating radar data and further increasing the accuracy of flood forecasting, although both precipitation and flood are slightly over-forecasted. However, the improvements by assimilating conventional observational data are not obvious. In general, radar data assimilation can improve flood forecasting effectively in a small-medium sized basin based on the atmospheric-hydrological coupling model.

**Keywords:** radar data assimilation; flood forecasting; atmospheric-hydrological coupling; WRF-Hydro model; Qingjiang River Basin

## 1. Introduction

Flooding is one of the most devastating natural disasters relative to human survival and development in the world [1–3]. In a country such as China where the topography is steep and the catchments in the mountains are prone to suffer from rainstorms, flood disasters occur frequently. Flood forecasting is one of the most important measures to mitigate the effects of floods on life and property [4,5].

The development of distributed hydrological models in the past decades has provided the potential to improve the watershed flood forecasting capability [6,7]. The watersheds are divided into different grids by distributed hydrological models. These grids have their own characters and are regarded to play the same role as that of a real watershed. With this feature, the inhomogeneity of both the terrain property and precipitation over watershed is well described. As a result, a better simulation of hydrological processes at any watershed scales can be obtained and the Quantitative Precipitation Forecast (QPF) from Numerical Weather Prediction (NWP) models can be better coupled as well [8]. The WRF-Hydro (Weather Research and Forecasting model (WRF) hydrological modeling system) model is

a physically-based and fully-distributed multi-parameterization modeling system that is easy to couple with NWP models and has the potential for operational flood forecasting [9]. The model has also been employed in many countries and achieved great simulation results [10–13].

Currently, the majority of hydrological models used for flood forecasting are driven by gauge rainfall data [14]. In this situation, the leading time depends on the catchment routing time [15]. For topographically steep catchments with extreme rainfall and flood, the leading time is too short, rendering it difficult to satisfy the requirements of early flood warning and emergency responses [16]. In addition, the gauge rainfall data are unavailable in the mountainous regions in general, rendering flood forecasting more difficult. At the same time, flood disasters occur frequently in those mountainous regions. Therefore, effective flood forecasting is needed urgently in those regions.

The high-resolution NWP models are able to provide QPF products in the data-scare regions mentioned above and can be coupled with hydrological models, which is an effective method for extending the leading time of flood forecasting [17,18]. However, the QPF from NWP models is not accurate enough at present, which will further affect the quality of the flood forecasting [19–21] especially in the mountainous areas. It is necessary to improve the quality of QPF before coupled to the hydrological model. Therefore, the bias-corrected QPF is adopted by many hydrological applications. However, due to the non-independence of the corrected variables, the application of bias-corrected QPF would result in physical inconsistencies, thus reducing the utility of the full dynamical simulation [22,23].

Data assimilation is proven to be an effective method for improving NWP, and it is widely used [24]. It can correct the initial state of the atmosphere in NWP models by absorbing multi-source observations, further resulting in better forecasting results. Many scientists have shown that the QPF can be improved by data assimilation and in turn resulting in more accurate flood forecast results [25,26]. There are many types of observation data that can be assimilated, such as conventional observations and weather radar data, etc. Conventional observations include land surface, marine surface, radiosonde and aircraft reports, etc., and are convenient for being assimilated due to the uncomplicated observation operators [27,28]. However, conventional observations are unable to observe the structure and characteristics of precipitation directly, and its coarse temporal and spatial resolution are hardly satisfies the requirements of small-scale flood forecasting.

In contrast, the Doppler weather radar can measure the differences between the high frequency of the received signal and the transmitting signal when the precipitation particles move relative to the radar transmitting beam and are able to obtain the required precipitation information [29–31]. Consequently, radar data, with its ability to observe the fine structure and characteristics of precipitation and its extremely high temporal and spatial resolution, show great potential to improve the QPF and further enhance the quality of flood forecasting in small-scale regions. Therefore, it is of great significance to apply radar data assimilation to the coupled atmospheric-hydrological model for flood forecasting.

The motivation of this study is to investigate the impacts of radar data assimilation on flood forecasting capability in a small-medium sized basin based on a high-resolution atmospheric-hydrological coupling model. Multiple and typical case studies were conducted in the Qingjiang River Basin in HuBei China in order to evaluate the flood forecasting capability of this coupled model, which could provide reference for near real-time flood forecasting. The full text structure is as follows: Section 2 introduces the methodology and data. The experimental design and the calibration and validation of the WRF-Hydro model are described in Sections 3 and 4, respectively. Section 5 evaluates the performance of the coupled model simulations with respect to the precipitation and streamflow. Discussions are presented in order to provide a summary of the outcomes in Section 6.

## 2. Radar Data Assimilation Methodologies and the Coupled Model

### 2.1. Radar Data Assimilation Method and Quality Control

#### 2.1.1. WRF-3DVar Data Assimilation

The 3DVar method is widely used in research communities and operational centers due to the fast and efficient analysis it provides, and it is used in this study to assimilate conventional observations and radar data. The 3DVar method solves the data assimilation problem by minimizing the following cost function [32,33]:

$$J(x) = \frac{1}{2}\left(x - x^b\right)^T \boldsymbol{B}^{-1}\left(x - x^b\right) + \frac{1}{2}(H(x) - y^o)^T \boldsymbol{R}^{-1}(H(x) - y^o) \tag{1}$$

where $x$ is the state variable, $x^b$ is the first guess, $y^o$ is the observation and H is the observation operator that transforms the analysis field from the model space to the observation space. The matrices $\boldsymbol{B}$ and $\boldsymbol{R}$ are the background and observation error covariance matrices, respectively. In practice, the matrix $\boldsymbol{B}$ is usually approximated with the climatological forecast error covariance matrix because the true background error covariance is unknown. The most widely used method for estimating the $\boldsymbol{B}$ matrix is the so-called NMC (National Meteorological Center, NCEP) method [34]. In this study, the NMC method was adopted to generate domain dependent $\boldsymbol{B}$ by taking differences between forecasts of 12 h and 24 h lengths valid at the same time in a month, which used a total of 60 samples [35].

#### 2.1.2. Radar Data Quality Control and Observation Operators

Both the radar radial velocity and reflectivity were assimilated in the WRF-3Dvar module at the same time. In order to remove the main sources of errors including ground clutter, signal attenuation, beam blocking and radio interference that may affect radar measurements, the measurements obtained by the weather radars were processed with a quality control algorithm before being assimilated. The quality control procedures include dealiasing radial velocities and removing non-meteorological echoes, ground clutter echoes and EMI (Electromagnetic Interference) echoes [36,37]. The observation error of the radar radial velocity is 2 m/s and that of the radar reflectivity is 2 dBZ [38,39].

The radial velocity ($V_r$) forward observation operator is calculated as follows [40]:

$$V_r = u\sin\varphi\cos\mu + v\cos\varphi\cos\mu + w\sin\mu \tag{2}$$

where $\mu$ is the elevation angle, $\varphi$ is the azimuth angle of radar beams, and $u$, $v$ and $w$ represent zonal, meridional and vertical velocities, respectively.

The indirect assimilation method was adopted for radar reflectivity data assimilation, which assimilates hydrometeor mixing ratios estimated from radar reflectivity [41,42]. The forward model for equivalent radar reflectivity factor ($Z_e$) is obtained by summing the contributions from three hydrometeor mixing ratios using the following formulation [43–45]:

$$Z_e = \begin{cases} Z(q_r), & T_b > 5\,°\mathrm{C} \\ Z(q_s) + Z(q_g), & T_b < -5\,°\mathrm{C} \\ \alpha Z(q_r) + (1-\alpha)\left[Z(q_s) + Z(q_g)\right], & -5\,°\mathrm{C} < T_b < 5\,°\mathrm{C} \end{cases} \tag{3}$$

where $Z(q_r)$, $Z(q_s)$ and $Z(q_g)$ are the reflectivity factors of rain, snow and graupel, respectively. $\alpha$ varies linearly between 0 at $T_b = -5\,°\mathrm{C}$ and 1 at $T_b = 5\,°\mathrm{C}$, and $T_b$ is the background temperature from the NWP model. The calculation of the equivalent reflectivity factors contributed by each species can be simplified to a $Z$-$q$ relation, which is expressed most generally as follows:

$$Z(q_x) = \beta_x(\rho q_x)^{1.75} \tag{4}$$

where $\rho$ is the air density, $q_x$ is the mixing ratio of hydrometeor species x (e.g., "r" for rain, "s" for snow or "g" for graupel) and $\beta_x$ is the coefficient determined by the dielectric factor, density and intercept parameter of hydrometeor x. As in previous studies, $\beta_x$ is frequently treated as a constant, where $\beta_r$ is $3.63 \times 10^9$ [46] and $\beta_g$ is $4.33 \times 10^{10}$ [45].

However, the coefficient is considered to be temperature dependent for snow: When the temperature is greater than 0 °C, the coefficient for wet snow $\beta_s$ is $4.26 \times 10^{11}$ [45]. For dry snow, which occurs at temperature less than 0 °C, $\beta_s$ is $9.80 \times 10^8$ [47]. The last step converts the equivalent reflectivity factor to the customary logarithmic scale (dBZ) by using the following.

$$Z_{dBZ} = 10\log_{10}Z_e \tag{5}$$

In addition, the indirect assimilation combines with an option that allows the assimilation of the in-cloud humidity estimated from reflectivity at the same time [48], and it was used in this study. The observation operator is defined by the following:

$$q_v = rh \times q_s \tag{6}$$

where $q_v$ is the specific humidity, rh is the relative humidity and $q_s$ is the saturated specific humidity of water vapor. Thus, the assimilation of the in-cloud humidity is also included in addition to the hydrometeor species retrieved with the indirect method alone.

### 2.2. The Coupling Flood Forecasting Methods Based on Radar Data Assimilation

The atmosphere-hydrological coupling model with data assimilation constructed in this paper is mainly composed of the WRF model (Version 4.1.1), the WRFDA system (Version 4.1.1) and the WRF-Hydro model (Version 5.0.3).

The WRF-Hydro model is a distributed physical land surface hydrological model based on grid points, designed to facilitate the land surface modeling coupling with WRF model [49]. It extends the traditional one-dimensional land surface parametrization in WRF by providing routing modules including subsurface flow routing, overland flow routing, a bucket model to account for baseflow and channel flow routing. The WRF-Hydro model is set up by using multiple grid structures. The Land Surface Model (LSM) works for soil infiltration and redistribution in the course grid. When the routing option is switched on, a fine resolution grid with the capability of resolving local topography will further redistribute terrestrial moisture. The WRF-Hydro model can be used both in an uncoupled mode and in a coupled mode. In uncoupled mode, it requires externally and spatially distributed meteorological forcings. In the coupled mode, the NWP model provides the required forcings for WRF-Hydro [50].

In this paper, the main WRF-Hydro model parameters were calibrated and validated in the uncoupled mode firstly before coupling with the WRF model. It was based on the hourly observational forcing data that were prepared externally, including rainfall, incoming longwave and shortwave radiation, surface pressure, 2 m specific humidity, 2 m air temperature and 10 m wind in gridded format, and the spin-up period of the WRF-Hydro model was set to two months for each event [51]. Then, in the coupled mode, the WRF model provided the required meteorological forcing data in the innermost domain for the WRF-Hydro model hourly in each experiment.

Figure 1 shows a sketch of the coupled atmospheric-hydrological system based on radar data assimilation. The atmospheric and hydrological coupling is one-way in this study for coupling flexibility and the stability of model integral, and there is no feedback between the WRF model and the WRF-Hydro model. Note that a two-month spin-up run driven by the observational meteorological forcings is performed in the WRF-Hydro model before each event, which is identical to the parameter calibration process.

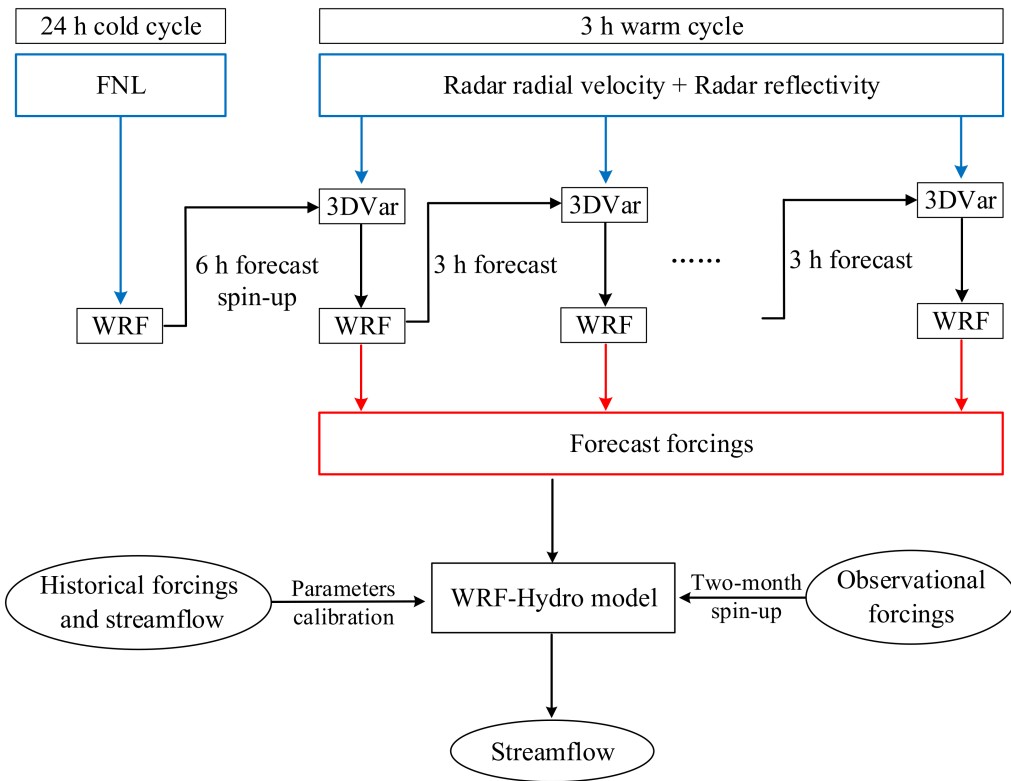

**Figure 1.** Flowchart of the coupled atmospheric-hydrological system based on radar data assimilation.

## 3. Experimental Setups of the Atmospheric-Hydrological Coupled Model

### 3.1. Qingjiang River Basin

The Qingjiang River Basin is located in HuBei province in middle China, which is the first-order tributary of the Yangtze River. Figure 2 is a sketch map of the Qingjiang River Basin. The basin covers a drainage area of 10,860 km², and the length of the main river is about 423 km. The Shuibuya hydrology gauging station is located at the basin outlet and the hourly streamflow data of the station are provided by the HuBei Key Laboratory for Heavy Rain Monitoring and Warning Research, Institute of Heavy Rain, China Meteorological Administration (CMA). According to the Moderate-Resolution Imaging Spectroradiometer (MODIS) 21-category Land Use Categories, vegetation types of the basin mainly include mixed forests (54%), woody savannas (19%) and deciduous needleleaf forests (10%) (Figure 2b). The soil category of the top layer is clay loam (Figure 2c).

The basin belongs to the subtropical monsoon region, where rainstorms with high intensity and short duration mainly occur in the summer. The annual rainfall distribution is extremely uneven, mainly concentrated from April to September accounting for 75–78% of the annual rainfall. The elevation of the basin varies between 362 m and 2233 m. The slope of the basin and the river channels is steep relatively; thus, the floods frequently occur with the short flood routing time. The extreme rainstorms easily induce severe flooding in the basin and cause huge flood damages [52].

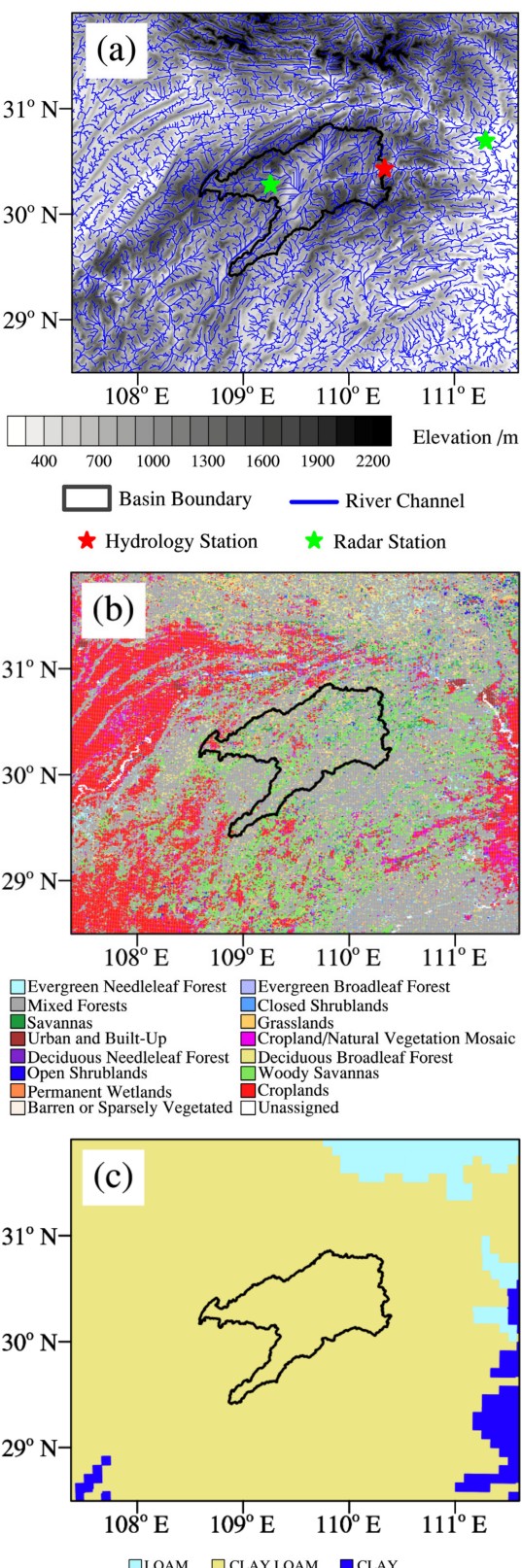

**Figure 2.** Study areas: (**a**) spatial distribution of elevation, basin, river channel, hydrology and radar station; (**b**) land use; (**c**) soil category.

### 3.2. The Coupled Model Configurations

#### 3.2.1. WRF Model and Data Assimilation Configurations

In this study, the dynamical core of the Advanced Research WRF is used. A three nested domain is set up with horizontal resolution of 15 km, 3 km and 1 km, respectively (Figure 3a). The vertical resolution in all the three domains is for 42 vertical levels with a model top at 50 hPa. The microphysical processes scheme is Thompson, the cumulus parametrization scheme is Kain–Fritsch (only used in d01), the land surface model is Noah, the longwave radiation scheme is RRTM, the shortwave radiation scheme is Goddard and the planetary boundary layer scheme is YSU [53,54]. The initial and boundary conditions of the WRF model are from the National Center for Environmental Prediction (NECP) and Final operational global analysis (FNL) with the resolutions of 0.25° and 6 h, respectively.

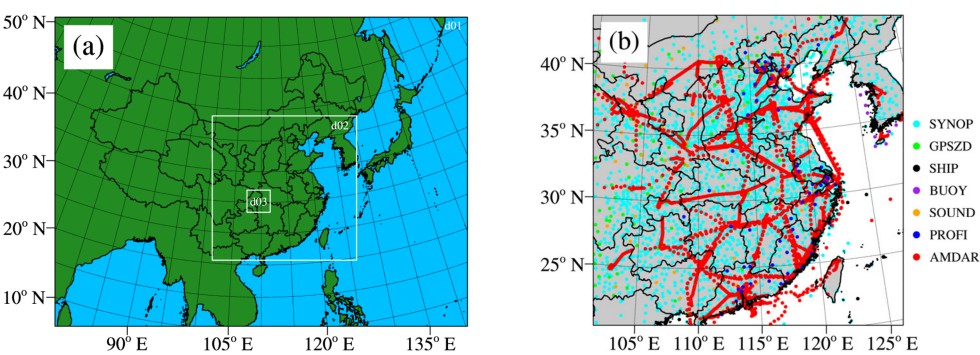

**Figure 3.** (**a**) The WRF domain and (**b**) the distributions of the GTS conventional observations assimilated in this study at 1200 UTC 7 July 2018 in WRF d02.

The data assimilated in the WRF model includes conventional observations and radar data. The conventional observations are from the Global Telecommunications System (GTS) conventional observational datasets in PREPBUFR format, provided by the NCEP with the time resolution of 6 h (Figure 3b). The radar data, including radial velocity and reflectivity in the S-band with the time resolution of 6 min and a maximum detection range of 230 km, are retrieved from the Yichang and Enshi station in HuBei Province. The detection range of the two radars covers the entire Qingjiang River Basin completely.

In order to examine how conventional observational data and radar data assimilation affect QPF, as well as flood forecasting, the following four experiments shown in Table 1 were mainly set up in this study. The GTS data were assimilated cyclically every 6 h based on CV5, and the radar data were assimilated cyclically every 3 h based on CV7 [55]. A 6 h spin-up run is conducted in the WRF model, and the output from the spin-up is used as the first guess field for the data assimilation. A cold start is performed every 24 h in all the four experiments. The data assimilation is performed in the WRF d02 with the resolution of 3 km. The time window of radar data is set to 15 min, and the time window of conventional observational data is set to 1 h.

**Table 1.** Experimental design.

| Experimental Name | Assimilation Scheme |
|---|---|
| CTRL | No assimilation |
| GTS | GTS conventional observations |
| RADAR | Radar radial velocity and reflectivity |
| GTS + RADAR | GTS conventional observations + Radar radial velocity and reflectivity |

#### 3.2.2. WRF-Hydro Model Configurations

The WRF-Hydro model was applied to WRF d03 (Figure 2). The coarse digital elevation and other ancillary data (e.g., land use and soil category) on LSM grids are generated

by the WPS (WRF Preprocessing System). The horizontal resolution of the coarse grid of the LSM is 1 km, which is identical to WRF d03. The fine horizontal resolution of the routing grids is set to 100 m by dividing the WRF d03 by a re-griding factor (10 in this paper). Note that the channel network was extracted from the fine-scale terrain data with stream definition threshold of 1000, thereby yielding a stream below every 0.01 km$^2$ of contributing area. The Noah-MP model is used as the LSM, the Steepest Descent method (D8) is used for the subsurface flow routing and overland flow routing, the exponential storage-discharge function is used for the baseflow model and the Diffusive Wave Formulation based on grid points is used for the channel flow routing [9].

The rainfall observational forcings are obtained from China Hourly Merged Precipitation Analysis (CHMPA), which is a gridded satellite-gauge-radar merged precipitation product with 0.05° resolution and developed by the National Meteorological Information Center of CMA [56]. The other observational meteorological forcing data including incoming longwave and shortwave radiation, surface pressure, 2 m specific humidity, 2 m air temperature and 10 m wind are from the ERA5 reanalysis datasets with hourly and 0.25° resolution, prepared by the European Centre for Medium-Range Weather Forecasts (ECMWF).

The Digital Elevation Model (DEM) data with a resolution of 90 m, which is required on routing grids, are provided by the Geospatial Data Cloud site, Computer Network Information Center, Chinese Academy of Sciences. The land use and soil category datasets with 30 s (second, 3600 s = 1°) resolution are retrieved from the MODIS datasets, all of them are processed by WRF-Hydro GIS Pre-Processing Tools in order to generate the input data of the WRF-Hydro model. The initial land surface state data of the WRF-Hydro model are all from the NECP FNL, with the spatial and temporal resolutions of 0.25° and 6 h, respectively.

## 4. Calibration and Validation of WRF-Hydro Model
### 4.1. Calibration Methods

In order to calibrate and validate the model parameters, thirteen typical flood events between 2015 and 2018 were adopted in this study, and their properties are shown in Table 2. The first ten flood events are used to calibrate the model parameters of the WRF-Hydro model, and the last three flood events are used for validation.

**Table 2.** The details of the thirteen flood events between 2015 and 2018.

| | Flood ID | Start Date | End Date | Peak Flow/m$^3$ s$^{-1}$ |
|---|---|---|---|---|
| | 20150531 | 12:00 31-05-2015 | 12:00 03-06-2015 | 2564.2 |
| | 20150629 | 12:00 29-06-2015 | 00:00 02-07-2015 | 2845 |
| | 20160624 | 00:00 24-06-2016 | 12:00 29-06-2016 | 4847 |
| | 20160630 | 00:00 30-06-2016 | 00:00 03-07-2016 | 5826.9 |
| Calibration | 20160718 | 00:00 18-07-2016 | 00:00 22-07-2016 | 12320 |
| | 20170609 | 00:00 09-06-2017 | 00:00 15-06-2017 | 2122.7 |
| | 20170707 | 12:00 07-07-2017 | 00:00 11-07-2017 | 4152.4 |
| | 20170713 | 12:00 13-07-2017 | 12:00 17-07-2017 | 4005.8 |
| | 20171001 | 00:00 01-10-2017 | 12:00 04-10-2017 | 6013.9 |
| | 20171011 | 00:00 11-10-2017 | 00:00 15-10-2017 | 2708.7 |
| | 20180505 | 00:00 05-05-2018 | 00:00 09-05-2018 | 2531 |
| Validation | 20180530 | 00:00 30-05-2018 | 00:00 02-06-2018 | 1409.1 |
| | 20180703 | 00:00 03-07-2018 | 00:00 08-07-2018 | 2079 |

The WRF-Hydro model contains many model parameters that have a great impact on the streamflow simulation results. In this study, four model parameters were calibrated, including referring soil permeability (REFKDT), multiplier on Manning's roughness for channel (MannN), multiplier on maximum retention depth (RETDEPRTFAC) and multiplier on Manning's roughness for overland flow (OVROUGHRTFAC) [57,58]. REFKDT is the most sensitive parameter relative to the streamflow simulation results, which determines the input flow in the channel confluence calculation. MannN and OVROUGHRTFAC

denote the roughness of the river channel and the surface land, respectively, and have remarkable effects on the speed of the water confluence. RETDEPRTFAC represents the water storage capacity of the surface land and needs to be adjusted according to the surface slope. The other model parameters used the model default values.

Considering the high computational cost, the Manual Stepwise Approach was adopted in this paper, which means driving the model with appropriate steps within a reasonable range to select the optimal parameter value according to objective function to calibrate the four main parameters in order [59]. In each step, we vary one parameter while keeping the other parameters constant. In this study, five kinds of objective functions in terms of correlation coefficient ($RR$), absolute value of the relative error of the total volumes ($|TE|$), absolute value of the relative error of the flood peak ($|PE|$), Nash-Sutcliffe efficiency coefficient ($NSE$) and absolute value of the flood peak time error ($|\Delta T|$) were used as calibration metrics. The details are as follows:

$$f_{RR} = \frac{1}{N} \sum_{n=1}^{n=N} \frac{\sum_{t=1}^{t=T} \left(Q_{obs}^t - \overline{Q_{obs}}\right) \left(Q_{sim}^t - \overline{Q_{sim}}\right)}{\sqrt{\sum_{t=1}^{t=T} \left(Q_{obs}^t - \overline{Q_{obs}}\right)^2} \sqrt{\sum_{t=1}^{t=T} \left(Q_{sim}^t - \overline{Q_{sim}}\right)^2}} \tag{7}$$

$$f_{|TE|} = \frac{1}{N} \sum_{n=1}^{n=N} \frac{\left| \sum_{t=1}^{t=T} Q_{obs}^t - \sum_{t=1}^{t=T} Q_{sim}^t \right|}{\sum_{t=1}^{t=T} Q_{obs}^t} \times 100\% \tag{8}$$

$$f_{|PE|} = \frac{1}{N} \sum_{n=1}^{n=N} \frac{\left| Q_{obs}^{max} - Q_{sim}^{max} \right|}{Q_{obs}^{max}} \times 100\% \tag{9}$$

$$f_{NSE} = \frac{1}{N} \sum_{n=1}^{n=N} \left(1 - \frac{\sum_{t=1}^{t=T} \left(Q_{obs}^t - Q_{sim}^t\right)^2}{\sum_{t=1}^{t=T} \left(Q_{obs}^t - \overline{Q_{obs}}\right)^2}\right) \tag{10}$$

$$f_{|\Delta T|} = \frac{1}{N} \sum_{n=1}^{n=N} \left| T^{Q_{obs}^{max}} - T^{Q_{sim}^{max}} \right| \tag{11}$$

where $f_{RR}$, $f_{|TE|}$, $f_{|PE|}$, $f_{NSE}$ and $f_{|\Delta T|}$ represent the objective functions for $RR$, $|TE|$, $|PE|$, $NSE$ and $|\Delta T|$, respectively. N represents the total number of the flood events, and T represents the total number of the time steps in each flood event. $Q_{obs}^t$ and $Q_{sim}^t$ represent the observed and simulated streamflow at time step t, respectively. $\overline{Q_{obs}}$ and $\overline{Q_{sim}}$ are the mean observed and simulated streamflow in each event. $Q_{obs}^{max}$ and $Q_{sim}^{max}$ are the observed and simulated peak streamflow in each event. $T^{Q_{obs}^{max}}$ and $T^{Q_{sim}^{max}}$ represent the flood peak time during an observed and simulated flood event, respectively. In each calibration step, the five metrics were considered comprehensively according to the physical meaning of the parameters in order to determine the optimal parameters value, and then the next parameter is calibrated while keeping the optimal parameters calibrated in the previous steps constant. The ranges and increments of the four parameters for the Manual Stepwise Approach are shown in Table 3.

**Table 3.** The ranges and increments of main four WRF-Hydro parameters for the Manual Stepwise Approach.

| Parameter | REFKDT | MannN | RETDEPRTFAC | OVROUGHRTFAC |
|---|---|---|---|---|
| Lower | 0 | 0.1 | 1 | 0.5 |
| Upper | 0.5 | 1 | 10 | 5 |
| Increment | 0.05 | 0.1 | 1 | 0.5 |

### 4.2. Results of Four Calibrated Parameters

Figure 4 shows the calibration process of the four parameters. Firstly, REFKDT mainly affects flood volume, so the optimal value was selected mainly based on the $|TE|$, $|PE|$ and $NSE$. It can be observed from Figure 4a that the $|TE|$ and $|PE|$ decreases with the in-

crease in REFKDT and begins to stabilize when REFKDT value equals to 0.1. While the *NSE* decreases as REFKDT increases, the optimal REFKDT value is set to 0.1. MannN chiefly affects the confluence speed of water in the river channels; therefore, the $|\Delta T|$ and *NSE* are comprehensively considered. According to Figure 4b, the MannN value 0.3 is taken to correspond to the smallest $|\Delta T|$ and the largest *NSE*. RETDEPRTFAC reflects the water storage capacity of the surface land. It can be observed from Figure 4c that the *NSE* and *RR*, which measure the overall performance of the simulated streamflow, do not change much with the increase in RETDEPRTFAC, indicating that the influence of RETDEPRTFAC on the streamflow simulation results in the river channels is not obvious. The RETDEPRT-FAC value 4 is chosen corresponding to the maximum *NSE* and *RR*. OVROUGHRTFAC measures the speed of water confluence on the surface land, so the $|\Delta T|$ and *NSE* are mainly considered. According to Figure 4d, the $|\Delta T|$ is basically stable within 3 h, and the optimal OVROUGHRTFAC value is set to 2.5 corresponding to the maximum *NSE*.

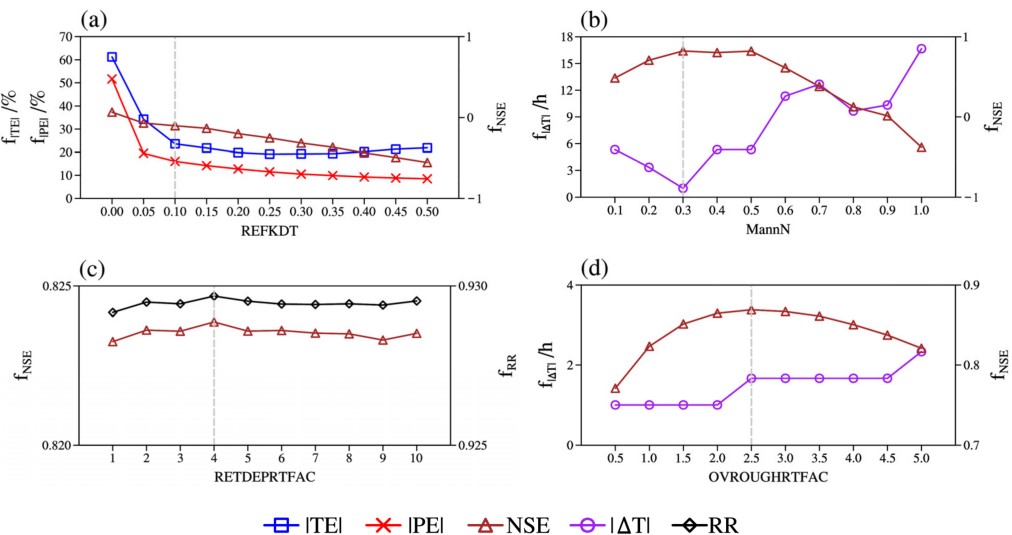

**Figure 4.** Calibration process of the four main model parameters (the gray dashed line represents the optimal value selected): (**a**) REFKDT; (**b**) MannN; (**c**) RETDEPRTFAC; (**d**) OVROUGHRTFAC.

According to the results of parameter calibration, it can be found that REFKDT and MannN obtained by calibration are relatively small. The main reason may be that the terrain slope is large. Consequently, the infiltration volume is marginal, and the velocity of channel flow routing is relatively fast. The simulation results are relatively less sensitive to RETDEPRTFAC. This may be due to the large variation of the altitude and the steep terrain in the study area, in which case there is little accumulation on the steep surfaces. Meanwhile, the soil is wet and close to saturation during the flood period, which further results in the little impact of RETDEPRTFAC on the streamflow simulation results in the study area. The OVROUGHRTFAC obtained by calibration is large relatively, which may be related to the strong interception effect due to the high forest coverage in the basin.

### 4.3. Validation of WRF-Hydro Model

Figure 5 shows the hourly streamflow simulation results of the thirteen flood events based on the optimal parameters. The rainfall in Figure 5 is the 1 h accumulated CHMPA, which is the average value of all the grids inside the Qingjiang River Basin (the same below). Overall, the observed streamflow corresponds well with the observed rainfall. It can be observed that the streamflow of the thirteen flood events simulated by the WRF-Hydro model is in great agreement with the observed streamflow as a whole. The multi-peak flood events (Event 20150629, Event 20160624, Event 20170609, Event 20170707, Event 20170713 and Event 20180703) are relatively complex, and the accumulation of the model errors due to the long duration of the events and the large catchment renders the streamflow

simulation results slightly worse than that of the single-peak flood events (Event 20150531, Event 20160630, Event 20160718, Event 20171001, Event 20171011, Event 20180505 and Event 20180530). The simulation deviation of the multi-peak events is mainly manifested in the simulation error of the flood peak intensity in Event 20170609, Event 20170713 and Event 20180703.

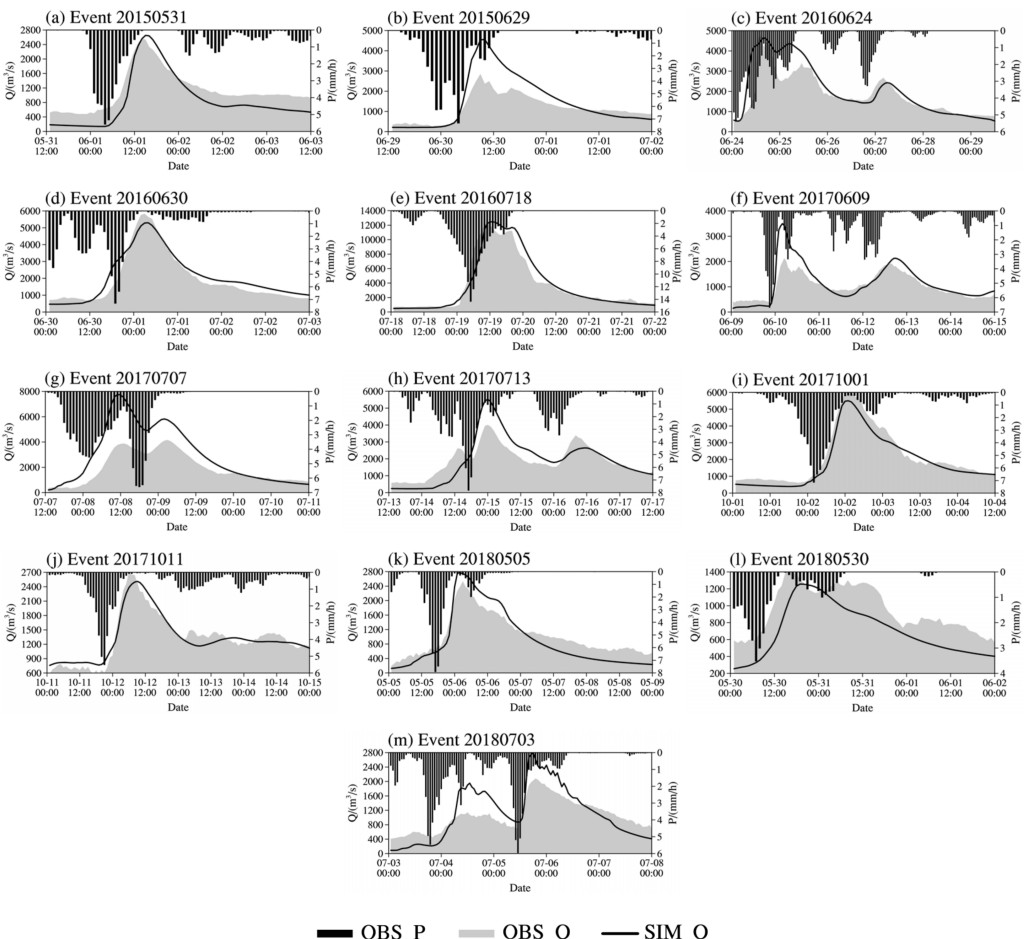

**Figure 5.** Hourly streamflow simulation results of the thirteen flood events. (**a**) Event 20150531; (**b**) Event 20150629; (**c**) Event 20160624; (**d**) Event 20160630; (**e**) Event 20160718; (**f**) Event 20170609; (**g**) Event 20170707; (**h**) Event 20170713; (**i**) Event 20171001; (**j**) Event 20171011; (**k**) Event 20180505; (**l**) Event 20180530; (**m**) Event 20180703.

The five metrics of the hourly simulated streamflow among the thirteen flood events are shown in Table 4. It can be observed that the *RR* of the thirteen events is generally above 0.9, indicating that the model can simulate the hydrographs well. The flood peak can also be grasped accurately by the model in most events. However, the *PE* of some individual events reaches more than 60%, which is mainly manifested in the simulation error of the flood peak of the multi-peak events of Event 20150629 and Event 20170609. The *NSE* of the thirteen events is between 0.50 and 0.94, which unfolds the high accuracy of the simulations. In addition, it was found that the simulation results of the single-peak flood events are better than that of the multi-peak flood events in general. The flood peak time error of all the events is basically within 3 h, indicating that the model captures the flood peak time well. There are two flood peaks in the Event 20170707 and their intensities are similar. Since the model incorrectly estimated the relative intensity of the two flood peaks, the flood peak time error is abnormal. The overall results show that the WRF-Hydro model after parameter calibration has strong applicability in the Qingjiang River Basin.

**Table 4.** The five metrics of the hourly simulated streamflow of the thirteen flood events.

|  | Flood ID | *RR* | *PE*/% | *TE*/% | *NSE* | *ΔT*/h |
|---|---|---|---|---|---|---|
| | 20150531 | 0.98 | 3.41 | −24.11 | 0.79 | 1 |
| | 20150629 | 0.96 | 60.31 | 27.30 | 0.69 | 1 |
| | 20160624 | 0.89 | −3.86 | 12.78 | 0.75 | 2 |
| | 20160630 | 0.97 | −9.04 | 7.68 | 0.92 | 1 |
| Calibration | 20160718 | 0.97 | 1.20 | 2.46 | 0.94 | 2 |
| | 20170609 | 0.91 | 64.89 | 11.55 | 0.73 | −1 |
| | 20170707 | 0.94 | 40.25 | 54.14 | 0.59 | −16 |
| | 20170713 | 0.89 | 38.10 | 2.83 | 0.74 | 0 |
| | 20171001 | 0.99 | −8.46 | −12.34 | 0.93 | −1 |
| | 20171011 | 0.96 | −7.24 | −0.26 | 0.91 | 3 |
| | 20180505 | 0.98 | 9.43 | −10.93 | 0.87 | −1 |
| Validation | 20180530 | 0.91 | −10.78 | −25.51 | 0.50 | 3 |
| | 20180703 | 0.91 | 34.42 | 6.92 | 0.71 | −1 |

In conclusion, with model parameters calibrated, the WRF-Hydro model performed well among different types of flood events. The seven single-peak flood events (Event 20150531, Event 20160630, Event 20160718, Event 20171001, Event 20171011, Event 20180505 and Event 20180530) are relatively simple, and the simulated streamflow is in great agreement with the observations. However, there are simulation deviations in the six multi-peak flood events (Event 20150629, Event 20160624, Event 20170609, Event 20170707, Event 20170713 and Event 20180703). The model error is mainly reflected in the simulation of the flood peak intensity in Event 20150629, Event 20170609 and Event 20170707.

## 5. Analysis of Coupling Forecast Results

In this section, the results of the four forecasting experiments under the atmospheric-hydrological coupling model are evaluated in order to examine the forecasting capability of the eight flood events between 2017 and 2018 since the complete meteorological and hydrological data of the eight events are available. The details of the eight events are shown in Table 2 (the last eight of the thirteen events).

### 5.1. Statistical Evaluation of Eight Flood Events

5.1.1. Evaluation of Rainfall

The QPF of the four experiments were compared with the CHMPA. Figure 6 shows the average TS and FSS of the hourly accumulated precipitation of the eight flood events, in which the radius of FSS is 5 km. According to the TS (Figure 6a), the overall 1 h QPF of RADAR experiment is well improved compared with CTRL and GTS experiments and passed the 95% significance test. The TS values of heavy rainfall (7.0 mm) and above are relatively low. In addition, the GTS experiment is slightly better than the CTRL experiment, but the improvement is not as great as RADAR and GTS + RADAR experiments. According to the FSS (Figure 6b), the results are basically consistent with the TS. The FSS of GTS, RADAR and GTS + RADAR experiments in each rainfall magnitude are all better than that of CTRL experiment, especially the RADAR experiment which shows distinctly improved precipitation.

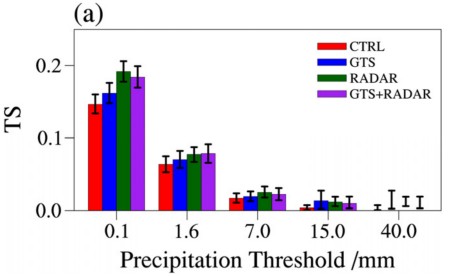 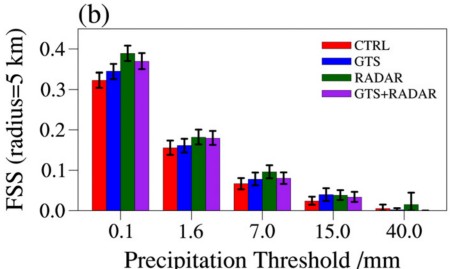

**Figure 6.** The average (**a**) Threat Score (TS) and (**b**) Fraction Skill Score (FSS) of the hourly accumulated precipitation of the eight flood events. (Error bar is 95% significance test interval.)

In order to further analyze the quality of the QPF, the correlation coefficient (*RR*), the time error of the peak ($\Delta T$), the relative error of the total volumes (*TE*) and the relative error of the peak volume (*PE*) of the areal rainfall in the Qingjiang River Basin were computed. Figure 7 shows the four metrics of the forecast areal rainfall among the eight flood events.

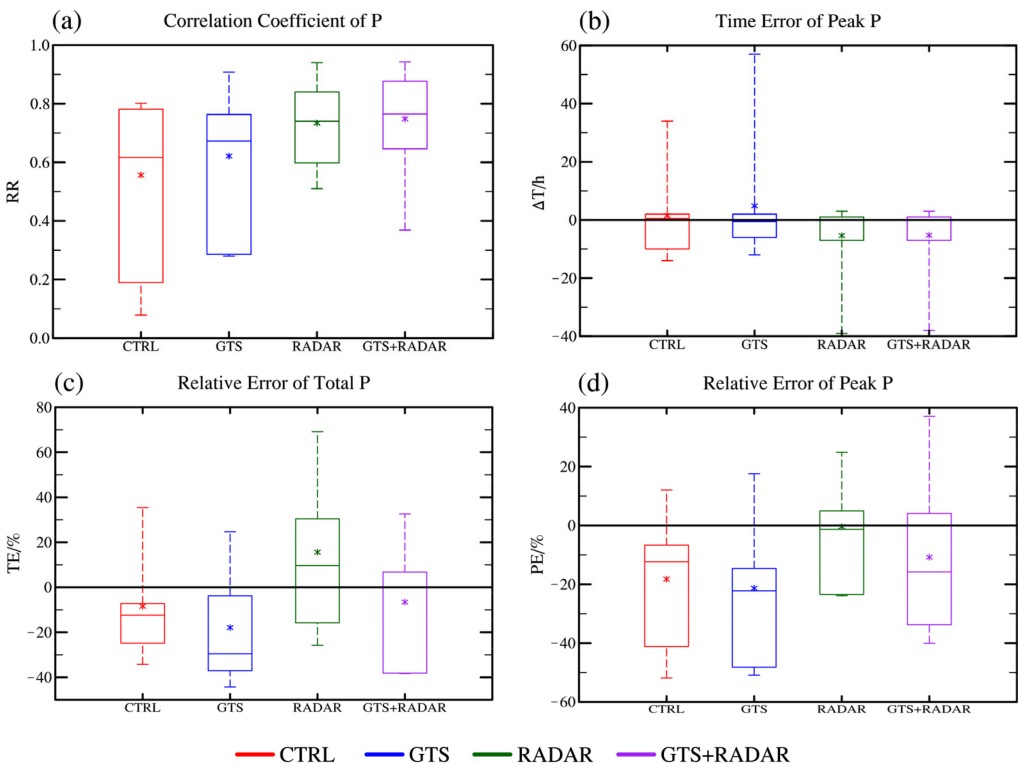

**Figure 7.** Boxplots of the evaluation metrics of the forecast areal rainfall in the Qingjiang River Basin of the eight flood events. (**a**) *RR*; (**b**) $\Delta T$; (**c**) *TE*; (**d**) *PE*. (* represents the average value.)

According to the *RR* (Figure 7a), all assimilation experiments have positive effects on the areal rainfall forecasts, especially RADAR and GTS + RADAR experiment which assimilated radar data. In addition, although the uncertainty between different events in CTRL experiment is relatively large, the *RR* is relatively high in general and the average value is above 0.5. After assimilating conventional observational data, the *RR* is further improved, and the uncertainty among different events is decreased. When radar data are assimilated, the *RR* is significantly improved, and the uncertainty is also distinctly decreased. In particular, assimilating radar data on the basis of assimilating conventional observational data performs the best. According to the $\Delta T$ (Figure 7b), GTS experiment reduces both the error and uncertainty compared with CTRL experiment, while that of RADAR and GTS + RADAR experiments are further reduced in general. The WRF model

incorrectly estimated the peak time of the areal rainfall during some individual cases, resulting in abnormally large errors. According to the *TE* (Figure 7c), all assimilation experiments increase uncertainty, and RADAR experiment overestimates the overall rainfall slightly. According to the *PE* (Figure 7d), both the peak areal rainfall of CTRL and GTS experiments are underestimated in most events. Compared with CTRL and GTS experiments, RADAR and GTS + RADAR experiments increase the peak value and reduce the peak error with the uncertainty decreased at the same time.

In general, compared with CTRL and GTS experiments, the assimilation of radar data (RADAR, GTS + RADAR experiments) renders the temporal distribution of the areal rainfall more accurate and the peak error smaller. Although the assimilation of the conventional observational data (GTS experiments) reduces the temporal distribution error of the areal rainfall, the uncertainty of the volume error is still relatively large, which may bring the overall forecasts negative impacts.

### 5.1.2. Evaluation of Streamflow

The quality of the streamflow forecast results under the coupled model in the four experiments were examined. Figure 8 shows the *NSE* of the simulated hydrographs of the eight flood events. It can be observed that the *NSE* of the simulated hydrographs driven by the observational forcing data (OBS experiment) is the highest, and the average value is above 0.7. In the CTRL experiment, the *NSE* of the optimal event among the eight events can reach 0.5, and the worst value is negative. After assimilating the conventional observational data, the *NSE* decreases and the uncertainty among the eight events increases slightly, indicating that the large-scale observational information has a great uncertainty relative to the impacts of the forecast results in small-scale region and sometimes may make the forecast results worse. After assimilating the radar data, the *NSE* significantly improved and the uncertainty greatly decreased. The average value of the *NSE* is 0.5 approximately with the optimal value close to 0.9, which is similar to the streamflow simulated by the observational forcing data. It is better to assimilate the radar data on the basis of assimilating the conventional observational data (GTS + RADAR experiment) compared with CTRL and GTS experiments, but the negative effects brought by the conventional observational data render the streamflow forecast results slightly worse than assimilating the radar data alone (RADAR experiment).

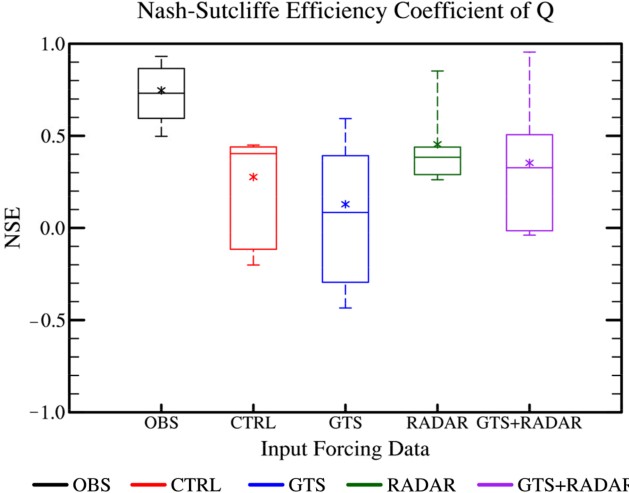

**Figure 8.** Boxplot of the Nash efficiency coefficient (*NSE*) of the simulated hydrographs of the eight flood events. (* represents the average value.)

In order to further evaluate the streamflow forecasts of the four experiments, the four evaluation metrics including the correlation coefficient (*RR*), the flood peak time error ($\Delta T$),

the relative error of the total volumes (*TE*) and the relative error of the flood peak volume (*PE*) of the simulated hydrographs of the eight flood events are shown in Figure 9.

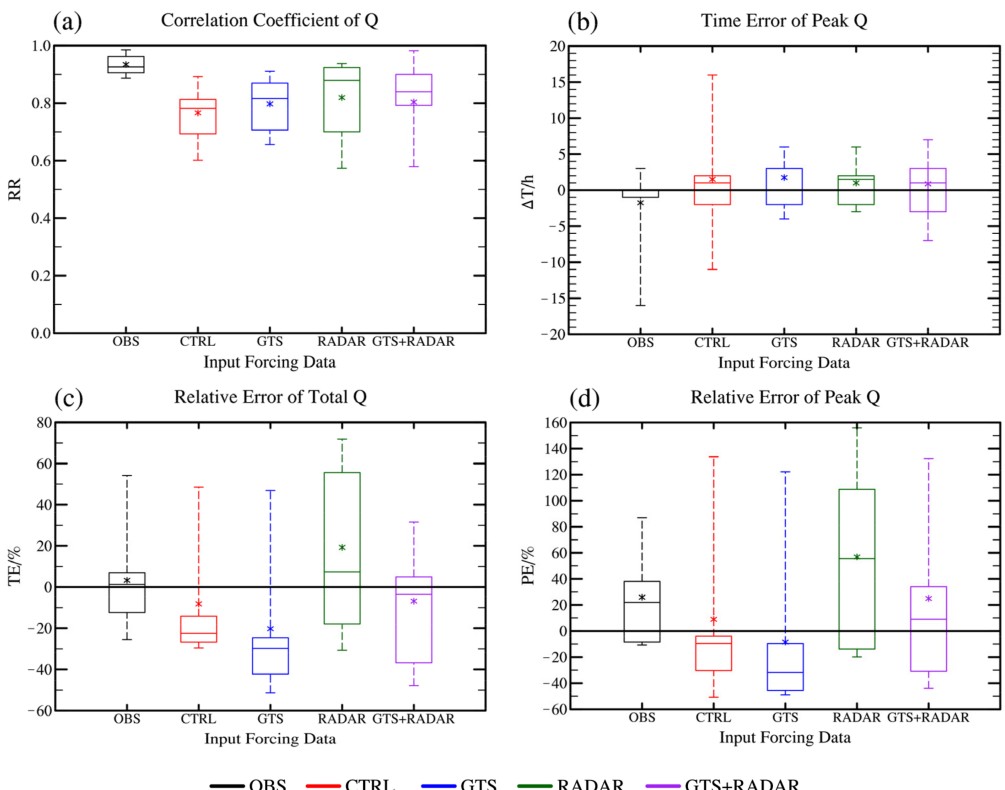

**Figure 9.** Boxplots of the evaluation metrics of the forecast hydrographs of the eight flood events. (**a**) *RR*; (**b**) Δ*T*; (**c**) *TE*; (**d**) *PE*. (* represents the average value.)

According to the *RR* (Figure 9a), the value of the streamflow simulated by the observational forcing data is the highest, and the uncertainty among different events is the smallest. The *RR*s in the four forecast experiments are all very high, and the average value is above 0.7, indicating that the model can predict the shape of the hydrographs well. After assimilating the conventional observational data, the overall *RR* is slightly higher than that in the CTRL experiment. The *RR* further improved but the uncertainty slightly increased in the RADAR experiment, while the uncertainty among different events decreased in the GTS + RADAR experiment compared with the RADAR experiment.

According to the Δ*T* (Figure 9b), the flood peak time simulated by the observational forcing data is generally accurate. However, there is an abnormality in some events due to the incorrect estimation of the flood peak. In the CTRL experiment, the Δ*T* is small, but the number of abnormal events increases compared with OBS experiment. In GTS experiment, the Δ*T* decreases, but the overall error increases slightly. The Δ*T* in RADAR experiment is slightly smaller than that in the CTRL experiment. The uncertainty of the Δ*T* in GTS + RADAR experiment is slightly increased with the overall Δ*T* within 5 h compared with that in the RADAR experiment.

According to the *TE* (Figure 9c), the *TE* of the streamflow simulated by the observational forcing data is overall between −10% and 10%, but there are also abnormal events where the *TE* exceeds 50%. The *TE* is generally small in the CTRL experiment and is smaller in the GTS experiment. Furthermore, after assimilating the radar data, the negative deviation of the total volumes improved in general, but the uncertainty increases. GTS + RADAR experiment performs best with smaller *TE* than the RADAR experiment, and the uncertainty also decreased.

According to the *PE* (Figure 9d), the flood peak volume in CTRL experiment is relatively small, and the *PE* is slightly larger after assimilating conventional observational data. The total volumes of some events are overestimated, and the uncertainty among different events increases in RADAR experiment. The *PE* in GTS + RADAR experiment is smaller than that in RADAR experiment, and the uncertainty is also decreased, which indicates the high predictability of the flood peak in GTS + RADAR experiment.

To sum up, it can be found that the assimilation of radar data (RADAR and GTS + RADAR experiments) improves the forecast hydrographs both in terms of shape and volume. However, there is also overestimation in flood volume, especially in RADAR experiment. With the assimilation of the conventional observational data, the shapes of the forecast streamflow are improved in general with both the total and peak flood volume slightly deteriorated. Therefore, while assimilating the conventional observational data, the *NSE*s are slightly smaller due to the volume error compared with the experiment without assimilation, which also reflects that the assimilation of the large-scale conventional observational information may to some extent have uncertainty for flood forecasting in small-scale regions.

### 5.2. Diagnosis of Different Flood Events

In this section, the coupling forecast results of two typical flood events, including a single-peak flood event and a multi-peak flood event, were analyzed in detail, with both precipitation and streamflow forecast results considered.

### 5.2.1. A Single-Peak Flood Event

Event 20171001 is a typical single-peak flood event. Figure 10 shows the forecast areal rainfall and streamflow in Event 20171001. According to the areal rainfall (Figure 10a–d), there is a peak in the observation with the highest intensity exceeding 6 mm. The predicted peak values in CTRL and GTS experiments are both slightly small with the values less than 6 mm. The peak value in CTRL experiment is smaller than that in the GTS experiment, and the value in the CTRL experiment is approximately 4 mm. After assimilating the radar data, the forecast areal rainfall peak is significantly enhanced. The predicted peaks in RADAR and GTS + RADAR experiments are close to the observation with the peak value both exceeding 8 mm, which is slightly larger than the observation.

According to the forecast streamflow (Figure 10e), the simulated streamflow in OBS experiment is in great agreement with the observation, indicating that the WRF-Hydro model with main model parameters calibrated could reasonably reproduce the flood event. Since the rainfall peak in CTRL and GTS experiments are both underestimated, the flood peaks in both two experiments failed to be forecasted accurately. The forecast flood peak values in the two experiments are both only about half of what is observed. Nevertheless, while assimilating the radar data, the forecast flood peak and the hydrograph are more accurate in both RADAR and GTS + RADAR experiments.

Figure 11 shows the spatial distribution of the elevation and streamflow in the Qingjiang River Basin at 1300 UTC 2 October 2017, which is the flood peak time in Event 20171001. Note that due to the non-availability of the observational data at all the river grid-cells, including tributaries, the evaluation of the streamflow forecast results in the four experiments was conducted based on the model results driven by the observational forcing data (OBS experiment). It can be observed that the simulated streamflow in OBS experiment is mainly distributed in the main river channel, and the closer to the downstream area, the larger the streamflow is. Meanwhile, the streamflow in the tributaries of the main river channel is also relatively large. The forecast results show that the spatial distributions of the streamflow in all the four forecast experiments are close to that in the OBS experiment, but all the forecast streamflow in the upper and middle reaches are weaker, especially in CTRL and GTS experiments. The streamflow in the main river channel and the tributaries in RADAR and GTS + RADAR experiments are generally consistent with that in OBS experiment, except that the streamflow in the upper reaches is relatively weak. The spatial

variability of the streamflow is well represented while assimilating the radar data in the coupled model. In summary, after assimilating the radar data, the spatial distribution of the forecast streamflow is effectively improved.

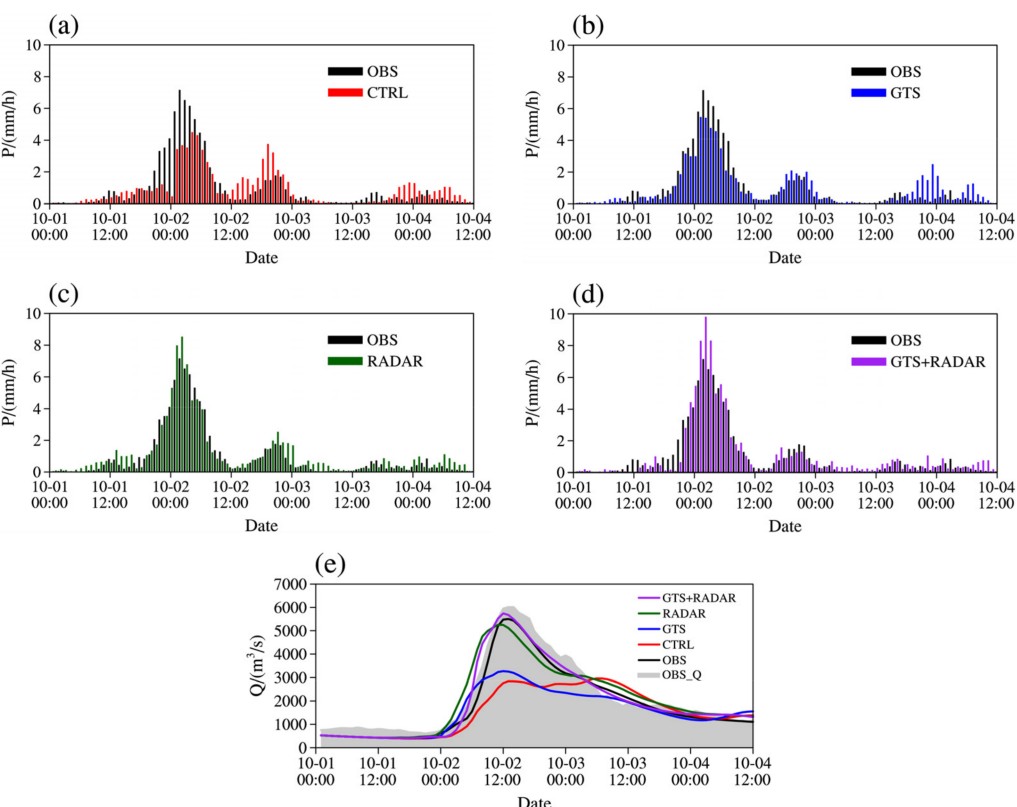

**Figure 10.** The forecast results of the areal rainfall in (**a**) CTRL, (**b**) GTS, (**c**) RADAR, (**d**) GTS + RADAR experiment and (**e**) streamflow in Event 20171001.

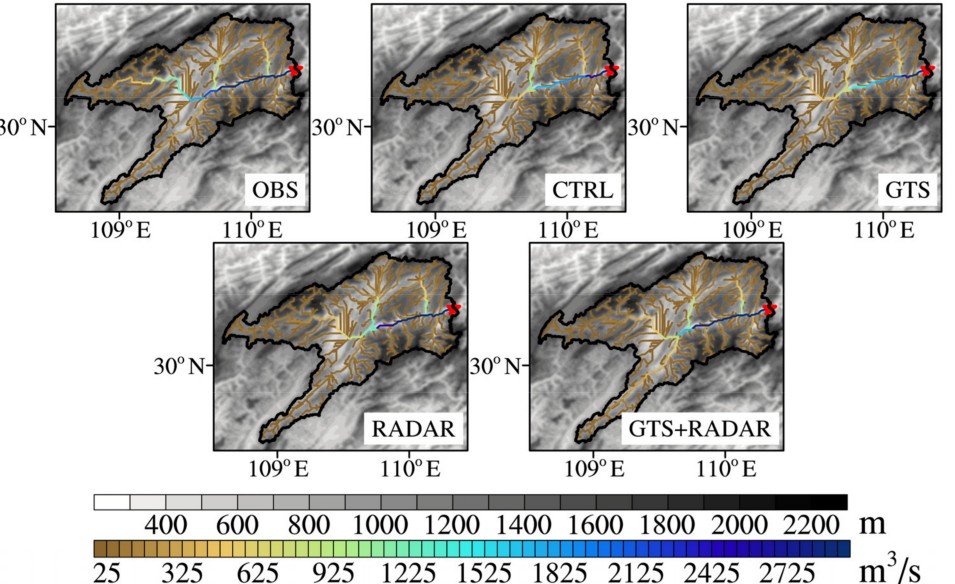

**Figure 11.** Spatial distribution of the elevation and streamflow in the Qingjiang River Basin at 1300 UTC 2 October 2017. (The red star represents the location of Shuibuya gauging station.)

5.2.2. A Multi-Peak Flood Event

Event 20170707 contains multiple flood peaks. Figure 12 shows the forecast areal rainfall and streamflow in Event 20170707. According to the areal rainfall (Figure 12a–d), there are two peaks in the observation. The first peak predicted in CTRL and GTS experiments is approximately 6 h late, and the forecast areal rainfall intensity in GTS experiment is exceptionally small. The first peak time predicted in RADAR and GTS + RADAR experiments is more accurate, but the peak intensity in GTS + RADAR experiment is too small. After the first peak, the forecast areal rainfalls in RADAR and GTS + RADAR experiments are both small. The second areal rainfall peak is predicted more accurately in CTRL and RADAR experiments than that in GTS and GTS + RADAR experiments, and the second peaks in GTS and GTS + RADAR experiments are late and small, indicating that the assimilation of the conventional observational data may bring negative impacts on the rainfall forecast in small-scale regions.

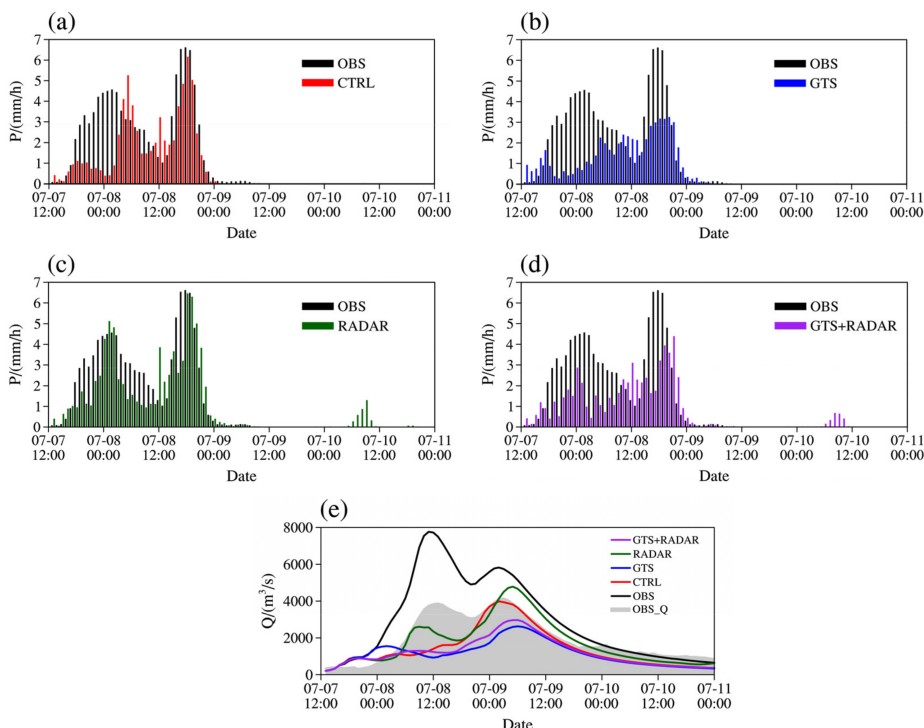

**Figure 12.** The forecast results of the areal rainfall in (**a**) CTRL, (**b**) GTS, (**c**) RADAR and (**d**) GTS + RADAR experiments and (**e**) streamflow in Event 20170707.

In terms of the forecast streamflow (Figure 12e), Event 20170707 contains two flood peaks. The streamflow in OBS experiment is generally consistent with the observed streamflow. However, there are distinct overestimations for the two flood peaks, indicating that there is a hydrological model error. The first flood peak is not predicted well in CTRL, GTS and GTS + RADAR experiments, which is partly attributed to the failure to predict the rainfall peak around 00:00 on 8 July. Although the first rainfall peak is predicted well in the RADAR experiment, the first forecast flood peak is ahead of time and the corresponding intensity is slightly weaker compared to the observation due to the underestimation of the previous rainfall. The second flood peak is overall well predicted in CTRL and RADAR experiments, and it is slightly late and stronger in the RADAR experiment due to the previous rainfall forecast. Due to the negative impacts brought by the conventional observational data in GTS and GTS + RADAR experiments, the second forecast flood peak is relatively weak. Since the previous rainfall is predicted insufficiently, even if the later rainfall forecast is accurate, the forecast flood peak is still small.

Figure 13 shows the spatial distribution of the elevation and streamflow in the Qingjiang River Basin at 1100 UTC 8 July 2017, which is the flood peak time in Event 20170707. It is found that the streamflow in OBS experiment is mainly distributed in the main river channel, and the closer to the downstream, the larger the streamflow is. In addition, the streamflow in the tributaries of the main river channel is also relatively large in OBS experiment. However, only the streamflow in the main river channel is relatively large in CTRL and GTS experiments, with the intensity weaker compared with OBS experiment. While assimilating the radar data alone (RADAR experiment), the spatial distribution of the streamflow is well improved, especially at the outlet of the basin where the intensity of the streamflow is generally consistent with that in OBS experiment. The streamflow in GTS + RADAR experiment is better than that in CTRL and GTS experiments, but worse than that in RADAR experiment. Therefore, the assimilation of the radar data can not only improve the forecast hydrographs at the outlet of the basin but also adjust the spatial distribution of the streamflow in the whole basin.

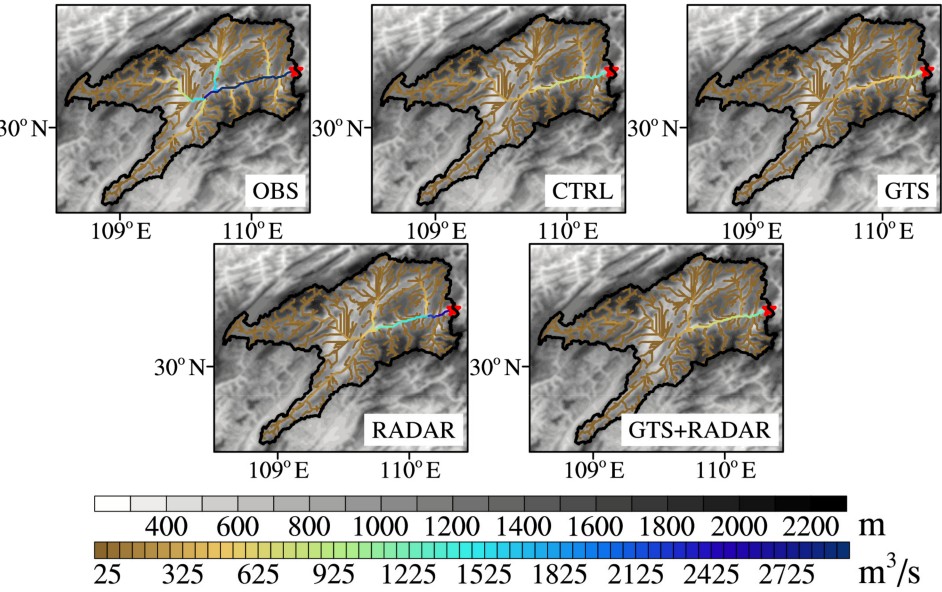

**Figure 13.** Spatial distribution of the elevation and streamflow in the Qingjiang River Basin at 1100 UTC 8 July 2017. (The red star represents the location of Shuibuya gauging station.)

In summary, the simulation error of the hydrological model also needs to be considered in the atmospheric-hydrological coupling forecasting. Meanwhile, when the forecast rainfall is used to drive the hydrological model for flood simulations, the rainfall forecast has a profound influence on flood forecasting. Therefore, even if the forecast rainfall peak is accurate, the previous rainfall forecast will play a significant role in the flood peak forecast. In addition, the assimilation of large-scale conventional observational data may have a negative impact on flood forecasting in small-scale regions, and the improvements of the flood forecasting mainly rely on the assimilation of unconventional high-resolution observational data such as weather radar datasets.

## 6. Discussion

Flood forecasting is important for mitigating the disasters caused by floods. It is difficult to accurately forecast flood when extending the flood forecast leading time by coupling the QPF products to a distributed hydrological model. Doppler weather radars can provide high-resolution observations, which show great indication for convective system, and have the potential to improve flood forecasting while assimilated in the atmospheric-hydrological coupling model. In this study, a high-resolution atmospheric-hydrological coupling model was constructed based on the WRF and WRF-Hydro models with conventional observational and radar data assimilation in a small-medium sized basin.

Eight typical cases study were conducted in the Qingjiang River Basin in order to evaluate the flood forecasting capability of the coupled model. Both precipitation and flood forecast results were statistically discussed, and two different types of flood events were analyzed in detail.

Coupling forecast results show that flood forecast results are highly dependent on the QPF in general. For precipitation forecast, the results with radar data assimilation are well improved according to TS and FSS, while the results with conventional observational data assimilation are marginally improved. Meanwhile, the temporal distribution and peak of areal rainfall with radar data assimilation are more accurate than no assimilation, although there is also overestimation. With conventional observational data assimilation, the temporal distribution of areal rainfall is more reasonable, but the amount is not obviously improved. For flood forecast, the *NSE* significantly improved when assimilating radar data due to the enhanced quality of the QPF, and the uncertainty of flood forecasts among different events also greatly decreased. After assimilating conventional observational data, the improvements of flood forecast results are marginal. In general, radar data assimilation can effectively improve flood forecasting based on the atmospheric-hydrological coupling model in a small-medium sized basin.

Due to the high temporal and spatial resolution of radar data, the forecast results in small-scale regions with radar data assimilation are generally improved in both precipitation and flood, although there is still over-prediction. In the future, more high-resolution observational data can be assimilated in NWP models, and more advanced data assimilation methods such as EnKF and hybrid methods for improving the overestimation can be adopted [60]. While assimilating conventional observational data, the improvements of the forecast results in both precipitation and flood are marginal. In addition, the uncertainty of the forecast quality among different flood events is still relatively large, especially in the flood volume error, which may bring negative impacts on the overall forecasts. It indicates that such large-scale observations may face difficulty in satisfying the requirements of small-scale flood forecasting.

In addition, the flood is sensitive to early rainfall. While early rainfall is not accurately predicted, the flood peak could not be reconstructed well even if the later rainfall is precisely predicted. This may be attributed to the inaccurate estimation of initial soil moisture in the flood event by the WRF-Hydro model. Therefore, we need to pay special attention to improve the prediction of the early rainfall in flood forecasting. For instance, experiments of different spin-up periods in the WRF-Hydro model can be conducted to improve the accuracy of initial soil water content in each flood event.

**Author Contributions:** Conceptualization, T.G. and Y.C.; methodology, T.G.; software, T.G. and Y.G.; validation, T.G., Y.C. and Y.G.; formal analysis, T.G. and Y.C.; data curation, T.G.; writing—original draft preparation, T.G.; writing—review and editing, T.G., Y.C., Y.G., L.Q., Y.W. (Yuqing Wu) and Y.W. (Yazhen Wu); visualization, T.G.; supervision, Y.C. and Y.G. All authors have read and agreed to the published version of the manuscript.

**Funding:** This work was jointly sponsored by the National Key Research and Development Program of China (2018YFC1506803) and National Natural Science Foundation of China (42075148).

**Data Availability Statement:** The open datasets used in this article are as follows. The NECP FNL were obtained from NCAR (https://rda.ucar.edu). The ECMWF ERA5 were obtained from ECMWF (https://cds.climate.copernicus.eu). The DEM data were obtained from Geospatial Data Cloud (http://www.gscloud.cn).

**Acknowledgments:** The numerical calculations in this paper have been performed on the super-computing system in the Supercomputing Center of Nanjing University of Information Science and Technology. The authors are grateful to the two anonymous reviewers for their constructive comments and useful suggestions that helped significantly improve the manuscript.

**Conflicts of Interest:** The authors declare no conflict of interest. The funders had no role in the design of the study; in the collection, analyses or interpretation of data; in the writing of the manuscript or in the decision to publish the results.

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
