# Peer review of "Improved Streamflow Forecast in a Small-Medium Sized River Basin with Coupled WRF and WRF-Hydro: Effects of Radar Data Assimilation"

_remotesensing, doi:10.3390/rs13163251_

Round 1

Reviewer 1 Report

General comments

This article discusses the impact of assimilating the radar observations into the WRF model to improve the QPE and streamflow forecast. This study used the WRF-Hydro model to predict the streamflow over Qingjiang River Basin, China. This paper was written well, and the results are discussed nicely. The findings from this paper are beneficial to other researchers in this field. Even though this paper is well written, it needs some improvements before publishing. I recommend accepting this paper after major revision.

Major comments

  1. Page 3, section 2.1.1. This section says the BE matrix is generated using the National Meteorological Center (NMC) method at the same time in a month. Please mention “domain dependent” BE matrix is generated and used. Would you please provide the number of ensembles used to create the BE matrix? Page 7, line number says CV5 was used to assimilate conventional data, and CV7 was used to assimilate the radar data. The author used gen_be version 2 utility, so please cite the related paper (Descombes, G et. al 2015).

  1. Page 3, section 2.1.2. Line number 116 says quality control done on data before assimilation. Please provide the details of the quality control procedure. At least discuss the outline.

  1. Page 3, Line number 117, Reference 32, (Gao and Stensrud 2012) is not mentioned about radial velocity forward observations. That paper is about the assimilation of hydrometeor mixing ratios. Would you please correct the citation? That paper should cite for Equ.3. Please cite the “Radhakrishnan and Chandrasekar 2020” paper, which followed the same assimilation method (assimilation of hydrometeor mixing ratios) in the WRF model.

  1. In page 3, section 2.1.2. Would you please provide the error distribution and standard deviation value used for radar observation (both reflectivity and radial velocity)?

  1. Page 5, figure 1. The warm start flowchart is not clear. Is WRF 3hr prediction used for the next WRF prediction cycle? Please check fig 2 in the “Radhakrishnan and Chandrasekar 2020” paper, and if possible, please try to plot a WRF & assimilation flow chart as shown in that paper. The figure is attached here.

  • Page 6, figure 2. Please show the radars location.

  • Page 7, line number 215, mentioned CV7 BE used for radar assimilation. Please refer the paper (Liu, C., Xue, M., & Kong, R. (2019)) that discusses hydrometeor mixing ratios background errors.

  • Page 7, line number 215, says the radar observations assimilated every 3 hours. Generally, radar observations are available in high temporal resolution (5 min). Would you please give justification for using 3hr cycle radar assimilation? 

  • Page 11, line number 364, says the details of the eight events are shown in Table 1. But the table 1 lists the four types of experiments. Please check and correct the statement if needed.

  • Recommend citing the papers from 4 to 7 (Gao 2017, Chandrasekar and Balaji (2015), Srinivas CV (2010,2012)). These papers are discussing the impact the assimilating the conventional observations.  

Reference suggestions:

  • Descombes, G., Auligné, T., Vandenberghe, F., Barker, D. M., and Barré, J.: Generalized background error covariance matrix model (GEN_BE v2.0), Geosci. Model Dev., 8, 669–696, https://doi.org/10.5194/gmd-8-669-2015, 2015.

  • Liu, C., Xue, M., & Kong, R. (2019). Direct Assimilation of Radar Reflectivity Data Using 3DVAR: Treatment of Hydrometeor Background Errors and OSSE Tests, Monthly Weather Review, 147(1), 17-29. Retrieved Jul 23, 2021, from https://journals.ametsoc.org/view/journals/mwre/147/1/mwr-d-18-0033.1.xml

  • Radhakrishnan, C., & Chandrasekar, V. (2020). CASA Prediction System over Dallas–Fort Worth Urban Network: Blending of Nowcasting and High-Resolution Numerical Weather Prediction Model, Journal of Atmospheric and Oceanic Technology, 37(2), 211-228. Retrieved Jul 23, 2021, from https://journals.ametsoc.org/view/journals/atot/37/2/jtech-d-18-0192.1.xml

  • Gao, S.; Huang, D. Assimilating Conventional and Doppler Radar Data with a Hybrid Approach to Improve Forecasting of a Convective System. Atmosphere 2017, 8, 188. https://doi.org/10.3390/atmos8100188

  • Chandrasekar, C. Balaji, Impact of physics parameterization and 3DVAR data assimilation on prediction of tropical cyclones in the Bay of Bengal region, Nat. Hazards (2015), pp. 1-25, 10.1007/s11069-015-1966-5

  • Srinivas CV, Yesubabu V, Hari Prasad K, Venkatraman B, Ramakrishna S (2012) Numerical simulation of cyclonic storms fanoos, nargis with assimilation of conventional and satellite observations using 3DVAR. Nat Hazards 63(2):867–889.

  • Srinivas, C.V., Yesubabu, V., Venkatesan, R. et al. Impact of assimilation of conventional and satellite meteorological observations on the numerical simulation of a Bay of Bengal Tropical Cyclone of November 2008 near Tamilnadu using WRF model. Meteorol Atmos Phys 110, 19–44 (2010). https://doi.org/10.1007/s00703-010-0102-z

Minor comments

  • Page 1, line number 12, “quantitative precipitation forecast (QPF)”. It should be It should be

            Quantitative Precipitation Forecast (QPF)”.

  • Page 1, line number 42 : numerical weather prediction(NWP). It should be “Numerical Weather Prediction”
  • Page 2, line number 56, Remove quantitative precipitation forecast, use QPE.
  • Page 5, line number 178:” km2 “ It should be km2
  • Page 10, line number 326, Event20170707, Put space between Event and date.

Reviewer 2 Report

The manuscript has sections where the grammar is very good and others where it is challenging. Improving the grammar will certainly help the reader's comprehension of the subject. 

The introduction outlines how an atmosphere-hydro coupled model will aid in early flood warning and emergency response.  Your experiment design does not seem to support this.  Using the NCEP FNL (late analysis) and ERA5 does not suggest your configuration could be run in near real time.

More detail is needed on how the models are coupled.  Is there any 2 way interaction?  How often is information from WRF used in WRF-Hydro? Every 3 hours?

Weather radars also have problems in mountainous regions. How did you account for the mountains in your data?  The paragraph starting at line 111 needs more information.

In line 202 you are using the NCEP FNL for boundary conditions.  Line 248 you are using ECMWF ERA5 reanalysis.  Mixing models is not a good idea. 

Figures 2a and 2b are too small to understand the elevation and the land use.  Similar is figure 11.  These are also too small to verify your point.

The paragraph starting at line 359 references table 1.  Is this correct?

Paragraph starting at 194 should have some model references.

Your study uses only flooding scenarios.  A false alarm or a missed flooding event are just as important for a real time forecast.  It would be helpful if you addressed the probabilities of the alternatives.  Otherwise, you just have case studies and is not very relevant to any near real time use.

Round 2

Reviewer 1 Report

I thank the authors for their responses to all my comments. The responses are very satisfying, I recommend accepting this article in the present form.

Author Response

Thanks again for your comments and suggestions offered to improve this manuscript. 

Reviewer 2 Report

Both the introduction and discussion need grammar improvements. 

line 34; "Flood forecasting is one of the most important measures for flood mitigation." This sentence implies that flood forecasting somehow modifies the actual weather.  The sentence should be something like "Flood forecasting is one of the most important measures to mitigate the effects of floods on life and property".  A similar edit is needed in the discussion.  It is correct in the abstract.

line 37; "The distributed ...." is a run-on sentence and needs to be revised.  I am not sure what point(s) the authors are trying to make. 

line 39; "This feature ..." another run-on sentence.

line 250;  "30 s resolution"  Do you mean "30 km square"?
